# Perception of the use of a telephone interpreting service during primary care consultations: A qualitative study with allophone migrants

**Maïmouna Jaiteh[1], Clément Cormi[1,2]\*, Louise Hannetel[3], Jean-Paul Mir[4], Edouard Leaune[5], Stéphane Sanchez[1,6]**

**1** Pôle Territorial Santé Publique et Performance des Hôpitaux Champagne Sud, Centre Hospitalier de Troyes, Troyes, France, **2** LIST3N/Tech-CICO, Troyes University of Technology, Troyes, France, **3** Palliative Care Unit, Centre Hospitalier de Troyes, Troyes, France, **4** Maison de Santé Pluriprofessionnelle, Communauté Professionnelle Territoriale de Santé du Sud-Est Aubois, Bar-sur-Seine, France, **5** Center for Suicide Prevention, Centre Hospitalier le Vinatier, Bron, France, **6** University Committee of Resources for Research in Health (CURRS), University of Reims Champagne-Ardenne, Reims, France

\* clement.cormi@ch-troyes.fr

**Data Availability Statement:** The minimal data set is within the manuscript (quote from allophone migrants interviewed) and the supporting information file 1 (interview guide). Data cannot be

# Abstract

## Objectives

The language barrier prevents allophone migrant patients from accessing healthcare when arriving in a country, and interpreters are often needed to help them to be understood. This study aimed to understand perceptions and experiences of allophone migrant patients who used a telephone-interpreting service during primary care consultations.

## Study design

A qualitative study using semi-structured interviews was undertaken between September 2019 and January 2020. Interviews were transcribed and analysed using thematic analysis framework.

## Setting

Allophone migrant patients from an accommodation centre for asylum-seekers who used a telephone-interpreting service during primary care consultations with a general practitioner.

## Participants

A purposive sample of allophone migrant patients (n = 10).

## Results

From the semi-structured interviews, we identified three themes: (1) multi-level difficulties of being an allophone migrant in the primary care pathway (i.e., before, during, and after the consultation); (2) the key role of the interpreter in the doctor-patient relationship, participating in improving the patient's management by establishing a climate of trust between the

shared publicly because of the French legislation (Délibération n° 2018-155 du 3 mai 2018 portant homologation de la méthodologie de référence relative aux traitements de données à caractère personnel mis en œuvre dans le cadre des recherches n'impliquant pas la personne humaine, des études et évaluations dans le domaine de la santé (MR-004)). Full data are available on reasonable request for researchers who meet the criteria for access to confidential data. Please contact Pôle Territorial Santé Publique et Performance, Centre Hospitalier de Troyes. Requests can be sent to Mr. Daniel Duserre, the Data Protection Officer of Troyes General Hospital, at [daniel.duserre@hcs-sante.fr] or to the co-author Dr. Snachez at [cht.direction@hcs-sante.fr].

**Funding:** Ten hours of telephone interpreting services for the purposes of this study were performed by the company ISM Interprétariat, and financed by the regional union of independent medical practitioners of the East of France (URPS Grand Est).

**Competing interests:** The authors have declared that no competing interests exist.

two; and (3) advantages and limitations of the TIS. However, even if a telephone-interpreting service is very helpful, allowing quick access to interpreters speaking the allophone patient's native language, certain situations would require the interpreter to see the patient to better guide the doctor during the consultation.

## Conclusion

Telephone-interpreting services enable improving communication and comprehension between allophone migrant patients and doctors. Nevertheless, the interpreter may sometimes need to physically see the patient to better guide the doctor. To do so, interpreting services using videoconferencing deserve wider development.

## Introduction

Among the European countries, France rates second (behind Germany) in terms of the number of asylum seekers entering the country [1]. The health status of migrants is often adversely affected by their vulnerable situation, and represents a major public health issue [2, 3]. The native language of many migrants is often not the language spoken in the country where they are arriving. These migrants, called "allophones", have cited the language barrier as the greatest obstacle to accessing healthcare in their host country [4, 5]. Indeed, the language barrier prevents allophone patients from acquiring healthcare knowledge [6]. Furthermore, the doctor-patient relationship is founded on a feeling of security and trust in the healthcare provider [4], and therefore, allophone patients are at risk of communication difficulties that can lessen their satisfaction with care [7]. In this regard, interpreters provide a much-needed solution to the language problem, whether they are formal, i.e. a trained professional interpreter, or informal (family members or friends of the patient) [8]. However, it has been reported that informal interpreting is not always satisfactory for allophone patients in primary care [9]. In particular, errors in translation may incur substantial clinical risk, and are significantly more frequent with informal interpreters than with trained interpreters [10]. Professional interpreters therefore have a major role to play during medical consultations, from an ethical point of view (to enable informed consent to be obtained), in improving the quality of care and follow-up (especially for chronic diseases, psychiatry or prevention) and in economic terms [11, 12].

Because of the scarcity of interpreting resources, telephone interpreting services (TIS) have become widespread since they first appeared in the 1970s [13]. While TIS present certain advantages, namely convenience, ease of access to qualified interpreters, and cost-effectiveness [14], some situations may not be amenable to the use of TIS. Wang et al. highlighted at least three such situations, all related to healthcare management, namely conversations with high emotional content, conversations about life or death, and medical settings (especially mental health consultations) [15]. Moreover, the accuracy of interpreting performance decreases by phone compared to face-to-face situations, while interpreters are more likely to make strategic additions (i.e., giving additional information to contribute to cultural understanding, effective communication and strong rapport between participants) [13, 16, 17]. TIS also face other challenges related to technical difficulties (e.g., poor sound quality), lack of training (e.g., managing three-way communication over the phone), and working conditions (e.g., low pay and isolation) [14]. According to Gracia-García, "a good interpreter at a distance is better than a bad one up close or none at all" [18]. Nevertheless, in a systematic review, Joseph et al. concluded that current evidence does not suggest that any one particular mode of interpreting is superior

to all others [19]. Given that TIS is a valuable service in a multicultural society, it deserves more attention in interpreter education, since it requires specific interpreting skills [20].

In France, the Regional Union of Independent Physicians in the Centre Loire region (URML des Pays de la Loire) has been providing interpreting services in the private-practice setting since 2017. It has also been providing interpreting services by telephone since 2018. On the basis of this experience, the Regional Union of Independent Medical Practitioners of the East of France (URPS Grand Est) decided to test a telephone-based interpreting service (TIS), initiated in October 2018. Physicians receiving allophone patients in consultation can contact the TIS free of charge, and be immediately put on to a professional interpreter. Two years after the introduction of this service, we sought to investigate how the allophone migrant patients perceive the use of this telephone-interpreting service during primary care consultations. As evidence from allophone migrant patients' perspective is missing [21, 22], the insights provided could help to better integrate this service into the practice of general practitioners (GPs) and primary care practitioners.

## Materials and methods

### Design and setting

We performed a qualitative study using semi-structured interviews with allophone patients who used the telephone-based interpreting service (TIS) during general practice consultations. The semi-structured interview gives the interviewee an opportunity to express their perception in their own terms, regarding the predefined topic [23].

### Study population

This was a single-centre study performed among allophone migrant patients who were resident at the time of the study in an accommodation centre for asylum-seekers (Centre d'Accueil de Demandeurs d'Asile, CADA) in Bar-sur-Seine, Eastern France. Allophone migrant patients who had at least one consultation in primary care using the TIS were eligible. Eligible patients (n = 32) were first identified by the GP at the primary care group practice in Bar-sur-Seine, and were then contacted by the social worker in the accommodation centre.

To account for the wide heterogeneity of allophone migrants likely to use the TIS, no inclusion criteria regarding age, sex, mother tongue, country of origin, socio-economic status or date of arrival in France were applied.

### Patient and public involvement

The study was motivated by informal but positive feedback from allophone migrant patients after using the TIS. However, they were not involved in the research design.

No formal feedback was organized because the migrants only stay a few months in the CADA. Most of the asylum seekers interviewed were no longer present at the time the results became available.

### Data collection

Each allophone migrant participated in a single semi-structured interview with a researcher (MJ: MD, GP, Primary Care researcher, female). Each interview began by calling the TIS to be connected with an interpreter who speaks the migrant's mother tongue. The first few minutes of the interview were dedicated to ensuring that the interviewee understood the study context, and that they could withdraw their consent at any time. The interpreter then asked the participant to confirm orally that they consented to participate.

Next, after discussing the circumstances of the participant's arrival in France, the interview focused on the participant's feedback concerning the use of the TIS during their medical consultation, and the impact of this service on the doctor-patient relationship. The main topics addressed were: how the patient came to France; how they learned about interpreting services; how they felt about the service, its usefulness and its impact on their healthcare experience. The full details of the interview guide are provided in the S1 Appendix.

With the oral consent of all the participants, the interviews were fully recorded. The recorder was placed in clear view on the table. Interviews were transcribed and rendered anonymous for later analysis.

## Data analysis

The data from the interviews were analysed using thematic analysis [24, 25]. Thematic analysis aims to identify and categorize the different themes occurring in a cross-sectional manner across all interviews. Each theme is a meaningful independent unit of the discourse. Major themes (relevant points, well developed by the participants) and secondary themes (themes of lesser importance, less well developed by the participants) are identified. Analysis was performed based on the semantic meaning of the sentences, and not the actual syntax, since this may be altered by translation.

Two rounds of analysis were performed independently by each of two researchers (MJ (female), LH (female)). The first round of analysis used open coding to identify the different themes present in the interviews, while the second round of analysis made it possible to classify these themes into major and minor themes, and to identify the relationships and hierarchies between them.

Interviews were performed until data saturation was reached (i.e. the point beyond which further interviews provide no new information) [26]. Data saturation was reached after nine interviews.

## Ethical considerations

In accordance with French legislation, Ethics Committee approval was not required for this study, in the absence of any intervention. This was confirmed by the Ethics Committee CCP Est I (Dijon, France). However, in view of the vulnerable status of allophone migrant populations, we ensured that all participants fully understood the objectives of the study and provided oral informed consent.

## Results

A total of 32 people were identified and contacted to participate. Thirteen of these agreed to participate, but 3 of them failed to show up at the agreed time for the interview. Therefore thus, from September 2019 to January 2020, a total of ten interviews were held (i.e. one with each participating allophone patient). The interviews were held in a dedicated room at the accommodation centre, which was known to all participants, and was quiet and informal.

The characteristics of the participants are detailed in Table 1. The average duration of the interviews was 34 minutes (range 23 to 63 minutes).

Data from the interviews produced eight thought clusters related to the allophone migrants' perception of the TIS. As shown in Table 2, these clusters were assembled to generate the following three major themes: (i) multi-level difficulties for allophone migrants in the healthcare pathway; (ii) the interpreter as the cornerstone of the doctor-patient relationship; and (iii) advantages and limitations of the TIS.

**Table 1. Characteristics of the participants.**

|  | Total |
|---|---|
| Age, mean (SD) | 32.7 (4.1) |
| Men, n | 6 |
| **Geographical origin, n** | **10** |
| Africa | 1 |
| America | 2 |
| Asia & Pacific | 1 |
| Central Europe & Asia | 2 |
| Middle East | 4 |
| **Time since arrival in France, n** | **10** |
| 3 to 6 months | 1 |
| 6 to 12 months | 2 |
| 12 to 24 months | 7 |

Being an allophone migrant: multi-level difficulties in the primary care pathwayOur findings revealed that the language barrier is present all along the migrant's consultation pathway, namely before, during and after the consultation.

Before the consultation, the obstacle of trying to get an appointment can make migrants delay, or abandon their recourse to care, via two mechanisms. Firstly, the language barrier can make it technically impossible to make an appointment, unless a third party can provide assistance: '*If I have a problem, or when I'm sick, the social worker has to make an appointment with the doctor*' (E2).

Second, the language barrier can prompt patients to fear that they will have difficulty making themselves understood during the consultation. Therefore, the patient will sometimes prefer to forego medical care rather than meet this difficulty head on, and the use of the TIS thus made it possible to get these allophone migrant patients in contact with the healthcare system at an earlier stage: '*I was in difficulty, but I preferred not to consult the doctor because I had trouble expressing myself, making myself understood [. . .] Now that there's an interpreter, I'm more inclined to make an appointment with the doctor*' (E4).

During the consultation, the interpreter is necessary, not to say indispensable in guaranteeing optimal management. This makes it possible for allophone patients to give detailed explanations about their motive for consulting the doctor:

**Table 2. Summary of the themes derived from the interviews and clustering of the allophone migrants' perception of the TIS.**

| Major Themes | Clusters | Subclusters |
|---|---|---|
| Being an allophone migrant is associated with multi-level difficulties in the primary care pathway | Before the consultation | Getting an appointment |
|  |  | Delay, give up |
|  | During the consultation | Optimal management |
|  |  | Bridge between the doctor and the patient |
|  | After the consultation | Therapeutic alliance |
| The interpreter as the cornerstone of the doctor-patient relationship | Alternative communication strategies | |
|  | Impersonal mode of communication | |
|  | Impact on doctor-patient relationship | Positive |
|  |  | Negative |
| Limitations of the TIS | The lack of an image | |
|  | A temporary solution | |

*'If there hadn't been an interpreter, we would have no use for a doctor' (E1). 'The interpreter can explain to the doctor what's wrong with me' (E3).*

This also helps to avoid errors in management: *'I tried to tell him I had a pain in my stomach, but I couldn't. I must've been misunderstood because the doctor prescribed me medication for heart problems' (E3).*

The interpreter acts as a bridge between the doctor and the patient, enabling flow of information in both directions. This mutual comprehension is crucial at the time of the consultation to promote the therapeutic alliance between the doctor and the patient: *'The interpreter is the tool that bridges the gap between me and the doctor, [. . .] the tool that enables me to communicate' (E7).*

After the consultation, this comprehension conditions the follow-up management and also contributes to patient compliance:

*'We understood each other much better [with the doctor]. He understood what I wanted, what I was asking as a patient' (E8).*

*'The interpreter ensures that the doctor doesn't get a wrong impression' (E7).*

## The interpreter as the cornerstone of a smooth doctor-patient relationship

When no professional interpreting service is available, allophone patients usually have two options for communicating with the doctor. First, the use of a third language, which is neither French nor the patient's mother tongue. This type of exchange is often in English, and usually rudimentary, if neither the patient nor the doctor has a good command of English: *'We tried speaking English with the doctor because he didn't understand Spanish' (E7).*

When the use of a third, intermediary language is not possible, then allophone patients may choose to be accompanied by an informal translator, someone they consider speaks sufficiently good French: *'The other asylum seekers who speak French, they helped him' (E4).*

Finally, the patient may choose a mixture of these two options, whereby an accompanying person speaks to the doctor in a third (intermediary) language on behalf of the patient: *'I had no choice but get help from my 12-year-old daughter who speaks a bit of English' (E10).*

Allophone migrants seeking medical care are often unaware that an interpreting system is available. Therefore, it is generally the physician who proposes it: *'When I arrived, [the doctor] asked me some questions, and asked me if I could speak French. I said no and then he proposed to call an interpreter' (E5).*

Calling on a professional interpreter helps to establish a climate of trust between the patient and the doctor: *'You need to be understood, you need to be able to trust' (E8).*

Access to an interpreter with the same cultural background is also reassuring for the patients: *'When there's a Bengali interpreter, it's reassuring' (E6).*

In the context of healthcare delivery to refugees, this trust is essential to enable the patient to speak freely and reveal motives for consultation that may be related to psychological disorders: *'When [the doctor] called the interpreter, it became easier. He saw that in me, there were things that went deeper than just the physical side' (E7).*

Overall, the interpreting service seemed to have a positive impact on the doctor-patient relationship: *'It makes the relationship with the doctor much easier' (E1).*

It is perceived as an impersonal mode of communication, and the participants in our study did not report any qualms about professional secrecy: *'The interpreter doesn't know your name or surname, he doesn't know you, he doesn't know who you are' (E8).*

*'I never even thought about the question of confidentiality. . . whether it bothers me or not' (E3).*

Two female interviewees mentioned the sex of the interpreter, as they had both consulted for gynaecological problems: '*For gynaecological consultations, for private things, I prefer it to be a woman' (E2)*.

However, for the other participants, the main thing was to be able to communicate with the doctor despite the language barrier: '*For me, it doesn't matter whether it's a man or a woman. The language barrier is more important than that minor detail' (E5)*.

## Advantages and limitations of the TIS

In most situations, the TIS was judged by the participants to be perfectly adequate: '*It doesn't matter if it's in person or by telephone, the main thing is to be understood' (E2)*.

This is all the more true in that the TIS presents the advantage of being rapidly accessible, and means that patients can access unplanned care without having to depend on an interpreter's availabilities: '*By phone, the advantage is that you have direct access at any time' (E6)*.

Conversely, in some situations, there is a need to see, in order to be able to complement the patient's oral explanations, and this need cannot be met by a telephone service: '*You can do it over the phone, but it's not the same when the person is present. They can see your suffering, you can show them where exactly the pain is' (E5)*.

In any case, although the participants reported that they would use the TIS again, they see it as a temporary solution, until such time as they have learned French: '*As long as I need them, I will use an interpreter, but I'm aiming to learn French so that in future, I don't need them anymore' (E7)*.

## Discussion

This study aimed to describe the perception of allophone migrant patients about the use of a telephone interpreting service (TIS) during a primary care consultation in France. Our results highlight the difficulties of allophone migrants throughout the primary care pathway (before, during, and after the consultation) and the role of the interpreter as the cornerstone of the doctor-patient relationship. However, the participants also pointed out some limitations of the use of TIS in this context.

The language barrier in healthcare has primarily been studied in the hospital setting, i.e. emergency admissions or during hospitalisation [7, 27, 28], with the focus often on an analysis of the costs versus the benefits [29, 30]. In order to avoid the emergency room becoming the only point of access to care for allophone migrants, ambulatory care in the community needs to be available to them, but few studies have investigated the utility of interpreting services in primary care. According to Bischoff and Denhaerynck, interpreting services help to optimize the patient's healthcare pathway by facilitating access to primary care and prevention, and by reducing use of emergency services and futile complementary examinations [31]. Nevertheless, their study focused on health care costs, not on patient satisfaction or patients' representations about the interpreting service. Our study provides complementary insights, since participants unanimously agreed that the TIS was useful, and helped to make the exchange of information easier, ensuring that the information was exhaustive for both parties. According to our results, TIS makes it possible to facilitate access to primary care for allophone patients.

Our results revealed that overcoming the language barrier also made it possible for allophone patients to address other motives for consultation, such as potential psychological disorders. By sharing a common language, or even a common dialect, with someone who comes from the same country, or even the same ethnic group, the migrant and the interpreter share certain cultural specificities that may have an influence on the patient's management [32, 33]. A systematic review by Bauer and Alegría about the impact of interpreting on the quality of

psychiatric care shows that professional interpreters improve disclosure and attenuate difficulties. Indeed, evaluation in a patient's non-primary language may lead to incomplete or distorted mental status assessment, and sometimes interpreting errors when using non-professional interpreters [34]. These pitfalls can be avoided by the use of professional TIS.

Although the different forms of translation and interpreting (e.g. telephone or video, versus in-person) seem to provide a similar level of satisfaction for users [19], our study shows that some patients would prefer to be able to add gestures to their oral explanations, so that the interpreter could get a more comprehensive overview of the situation. TIS and face-to-face interpreters seem to be complementary, as both services meet different needs among patients. One solution that combines the rapid response of the TIS with the images and non-verbal information of the presence-based situation, while simultaneously limiting costs, would be to contact the interpreters by video-conferencing [11]. Recent work by Krystallidou shows that the gaze, gestures, and orientation of the body of participants are valuable resources for interpreters to help them guide interactions between participants [35]. The TIS used in our study performed 156 interpreting services via video-conferencing compared to 330,000 telephone interpreting sessions in 2020 [36], so there is a room for increased use of the video component. In our study, the participants had a positive view of the fact that they could not be identified by the interpreter. René de Cotret and colleagues have made recommendations for developing videoconference-based remote interpreting [37]. As it is spreading, additional studies seem warranted to identify the place that video-interpreting should take in primary care for allophone migrants.

Rather than involving interpreters, Schulz et al., proposed to organize tele-consultations with doctors who speak the patient's mother tongue [38]. However, this solution might be challenging to implement because firstly, it is unlikely that a pool of physicians could be constituted to cover all the languages represented among the migrants, and across all the medical disciplines; and secondly, the patients are located in a specific geographical area and are usually in contact with healthcare providers in that area. Delegating the follow-up of these patients to other specialists who might be several hundred kilometres away, could lead to a breakdown in the healthcare trajectory for some patients. Therefore, further development of video-conference solutions for interpreting appears to be the most attractive solution at present.

## Study strengths and limitations

Few studies have investigated the patient's perspective of using TIS to facilitate access to healthcare in a host country where they do not speak the local language. This study therefore provides new insights into the utility of this service, from the point of view of the users. In addition, appropriate qualitative methods were used to ensure an accurate portrayal of the experience and perceptions across a diverse range of patients. Our study also has some limitations. Firstly, there is potential for bias due to the fact that the interviews were performed through an interpreter. Secondly, we interviewed only allophone migrants living in a shelter for asylum seekers, limiting the generalizability of our findings. Indeed, the results may not be applicable to countries with more established migrant communities. However, this choice helped reassure our participants and put them at ease in familiar surroundings, thus encouraging more open discussion. In addition, the participants in this study seem to be among those who have the greatest need of this type of service.

## Conclusion

Our study reports the perceptions of allophone migrant patients who used telephone interpreting services during primary care in France. We found that TIS enabled improved bidirectional

communication and comprehension, reinforcing patient empowerment. The interpreter facilitated the creation of a climate of trust between the doctor and the allophone patient, making it possible to reveal under-addressed reasons for consultation (such as psychological disorders). Although the immediate availability of an interpreter by telephone enables improved communication during the consultation, there may be a need to actually physically see the patient in some cases, in order to complement the oral discourse. Interpreting services using video-conferencing (image plus voice) deserve wider development, as a means to combine rapid availability of interpreting, with images and visual interaction, while simultaneously limiting costs.

## Supporting information

**S1 Appendix. Interview guide.**
(DOCX)

**S1 Table. Consolidated criteria for reporting qualitative research (COREQ).**
(DOCX)

## Acknowledgments

The authors wish to thank all the patients who participated in the study, as well as all the staff and management at the CADA shelter in Bar-sur-Sein for their availability during the study.

## Author Contributions

**Conceptualization:** Maïmouna Jaiteh, Louise Hannetel, Jean-Paul Mir, Stéphane Sanchez.

**Formal analysis:** Maïmouna Jaiteh, Clément Cormi, Louise Hannetel, Edouard Leaune.

**Investigation:** Maïmouna Jaiteh.

**Supervision:** Louise Hannetel, Jean-Paul Mir, Stéphane Sanchez.

**Writing – original draft:** Maïmouna Jaiteh, Clément Cormi, Louise Hannetel, Jean-Paul Mir, Edouard Leaune, Stéphane Sanchez.

**Writing – review & editing:** Maïmouna Jaiteh, Clément Cormi, Louise Hannetel, Jean-Paul Mir, Edouard Leaune, Stéphane Sanchez.

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
