## [Decision Letter · Decision Letter 0]

30 Sep 2021

PONE-D-21-18663Perception of the use of a telephone interpreting service during primary care consultations: A qualitative study with allophone migrantsPLOS ONE

Dear Dr. CORMI,

Thank you for submitting your manuscript to PLOS ONE. After careful consideration, we feel that it has merit but does not fully meet PLOS ONE’s publication criteria as it currently stands. Therefore, we invite you to submit a revised version of the manuscript that addresses the points raised during the review process.

We look forward to receiving your revised manuscript.

Kind regards,

Barbara Schouten

Academic Editor

PLOS ONE

Journal Requirements:

Reviewers' comments:

Reviewer's Responses to Questions

**Comments to the Author**

1. Is the manuscript technically sound, and do the data support the conclusions?

Reviewer #1: Yes

Reviewer #2: Partly

2. Has the statistical analysis been performed appropriately and rigorously? 

Reviewer #1: N/A

Reviewer #2: Yes

3. Have the authors made all data underlying the findings in their manuscript fully available?

Reviewer #1: Yes

Reviewer #2: No

4. Is the manuscript presented in an intelligible fashion and written in standard English?

Reviewer #1: Yes

Reviewer #2: No

5. Review Comments to the Author

Reviewer #1: Thank you very much for a very interesting piece of research. I miss a description of the themes heading each of the corresponding sections.

Further references about video interpreting in healthcare settins are the contributions by de Boe and by Krystallidou in Saalets, Heidi & Gert Brône (eds.) 2020. Linking up with video. Perspectives on interpreting practice and research. Amsterdam: John Benjamins.

Reviewer #2: The research paper, number PONE-D-21-18663, falls into the field of health interpreting and deals with a question of interest, yet little explored, that of patients' perception of telephone interpreting. However, the manuscript requires a stronger background to justify the research.

The introduction is indeed rather weak, as it lacks depth and articulation. In the first paragraph, the authors briefly address language barriers and their implications, the need for an interpreter and the two broad categories of interpreter (informal vs. professional). They directly present in the second (and last) paragraph the context of TIS in France and the contextual elements that led to the research. There is no transition in between. The different interpreting techniques and their benefits are missing (see Wang & Fang (2019) for a comparison between telephone interpreting and in-person interpreting in terms of translation quality), as well as the scientific data on TIS. There are at least two systematic review available (Downes, Mervin, Byrnes & Scuffham, 2017; Joseph, Garruba, & Melder, 2018). Lee (2007) and Wang (2018) have also investigated in the perception of interpreters. In addition, the last sentence of the first paragraph needs clarification. What do the authors mean by "qualitative terms" and "economic terms"?

The manuscript would gain in clarity, by reorganizing the structure and content of the Materials and methods section. The authors provided details regarding the population throughout the Method and at the beginning of Result section, while they should all be indicated in the same sub-section. The authors do not present the general topics explored in the interview (referring directly to the Supplementary Material is not sufficient). The sub-section “Telephone interpreting service (TIS)” does not appear necessary. Funding information are already given at the end of the manuscript and details about ISM can directly be included in the introduction. Please group information and be more straightforward.

Results are presented as a series of quotes, which does not do justice to the work involved by a qualitative analysis. The discussion, in its present state, does not bring anything new. The literature has already underlined the implication language barriers can have in primary care, and in mental health. Bauer & Alegria (2010) have for example conducted a literature review on interpreting in mental health.

Language polish is also strongly recommended. There are a number of linguistic typos throughout the manuscript. In the abstract, for example, the setting is presented as being the population, and not the accommodation center. I do not understand the label of the first theme "multiple translations of the language barrier". Can language barriers be translated?

Finally,

• Regarding "Ethics statement": Given the vulnerable status of asylum seekers, did the authors make sure that their research did not fall under a CPP, according to the Jardé Law as revised in 2016?

• Regarding "Data Availability": The authors indicate that there are restrictions on data accessibility but do not specify which ones.

• Regarding the COREQs: I suggest indicating directly information in the table, rather than indicating where in the text it can be found.

• Regarding Vancouver editing reference style: In the text, please use [], not ().

References cited in the review:

Bauer, A.M. & Alegria, M. (2010). Impact of patient language proficiency and interpreter service use on the quality of psychiatric care: a systematic review. Psychiatric Services, 61, 765–773.

Downes, M.J., Mervin, M.C., Byrnes, J.M., & Scuffham, P.A. (2017). Telephone consultations for general practice: a systematic review. Systematic Reviews, 6(1), 128.

Joseph, C., Garruba, M., & Melder, A. (2018). Patient satisfaction of telephone or video interpreter services compared with in-person services: a systematic review. Australian Health Review, 2(2):168-177.

Lee, J. (2007). Telephone interpreting, seen from the interpreters’ perspective. Interpreting, 6(2), 231–252.

Wang, J. & Fang, J (2019). Accuracy in telephone interpreting and on-site interpreting. A comparative study. Interpreting, 21(1), 36–61.

Wang, J. (2018). “It keeps me on my toes”, Interpreters’ perceptions of challenges in telephone interpreting and their coping strategies. Target, 30(3), 439–473.

6. PLOS authors have the option to publish the peer review history of their article (what does this mean?). If published, this will include your full peer review and any attached files.

Reviewer #1: **Yes: **Raquel Lázaro Gutiérrez

Reviewer #2: No

---

## [Author Response · Author response to Decision Letter 0]

12 Nov 2021

Thank you very much for your feedback on our work. Please find our responses to the reviewers’ comments in the 'Response to Reviewers' file.

---

## [Decision Letter · Decision Letter 1]

25 Jan 2022

PONE-D-21-18663R1Perception of the use of a telephone interpreting service during primary care consultations: A qualitative study with allophone migrantsPLOS ONE

Dear Dr. CORMI,

Thank you for submitting your manuscript to PLOS ONE. After careful consideration, we feel that it has merit but does not fully meet PLOS ONE’s publication criteria as it currently stands. Therefore, we invite you to submit a revised version of the manuscript that addresses the points raised during the review process.

We look forward to receiving your revised manuscript.

Kind regards,

Barbara Schouten

Academic Editor

PLOS ONE

Journal Requirements:

Reviewers' comments:

Reviewer's Responses to Questions

**Comments to the Author**

1. If the authors have adequately addressed your comments raised in a previous round of review and you feel that this manuscript is now acceptable for publication, you may indicate that here to bypass the “Comments to the Author” section, enter your conflict of interest statement in the “Confidential to Editor” section, and submit your "Accept" recommendation.

Reviewer #1: All comments have been addressed

Reviewer #2: All comments have been addressed

2. Is the manuscript technically sound, and do the data support the conclusions?

Reviewer #1: Yes

Reviewer #2: Yes

3. Has the statistical analysis been performed appropriately and rigorously? 

Reviewer #1: Yes

Reviewer #2: Yes

4. Have the authors made all data underlying the findings in their manuscript fully available?

Reviewer #1: Yes

Reviewer #2: Yes

5. Is the manuscript presented in an intelligible fashion and written in standard English?

Reviewer #1: Yes

Reviewer #2: Yes

6. Review Comments to the Author

Reviewer #1: (No Response)

Reviewer #2: Thank you to the authors for responding to my comments and the revisions they have made. The manuscript has gained in quality, and I have enjoyed reading this new version. I have a few more, minor comments or questions.

I came across a recent article on TIS over the Christmas vacation, which I believe would back up even more the introduction. It is not referenced in PsycInfo or PubMed, which is why -I’m guessing- neither the authors nor I have seen it before. It is, however, accessible on Researchgate upon request.

 René de Cotret, F., Beaudoin-Julien, A.-A., & Leanza, Y. (2020). Implementing and managing remote public service interpreting in response to COVID-19 and other challenges of globalization. Meta LXV, 3.

Two other articles from Leanza’s research team also seem relevant for the discussion.

 For the relational dynamic in interpreted consultation, including issue of trust (cf. lines 262-265) : Brisset, C., Leanza, Y. & Laforest, K. (2013). Working with interpreters in health care: A systematic review and meta-ethnography of qualitative studies. Patient Education and Counseling, 91, 131-140.

 A study conducted in primary care (cf. lines 312-314): Brisset, C., Leanza, Y., Rosenberg, E. et al. (2014). Language barriers in mental health care: A survey of primary care practitioners. Journal of immigrant and minority health, 16(6), 1238-1246.

More specifically,

* Line 44: The authors indicate two themes, while presenting three in the result section.

* Lines 85-86: The authors indicate that interpreters make more strategic additions during TIS. Although I find this results a surprising (and I will not argue on it), the complete sentence seems to me shaky. Strategic additions are a positive thing while the beginning of the sentence (and the following one) emphasizes on negative aspect of TIS.

* Lines 90-93: Would the authors consider moving the connector “Nevertheless” to the beginning of the next sentence?

* Lines 110-113: I wonder whether it is relevant (or not) to mention that the authors have followed the IMRAD structure, as the conducted research is prospective study.

* Lines 135-149: I first wondered here if the authors were able to ensure that it was a different interpreter than the one(s) used during the TIS consultations, which led me to wonder if patients using TIS are actually informed of who the person interpreting for them is…

* Line 184: Is it pertinent to add the %? I know it is a statistical standard but the sample relies on 10 participants…

* Lines 199-200: For table 2, would the authors consider presenting a figure instead of a table? I also found two typos: a lower-case s at “subclusters” instead of a capital S; and there is a lost ”;” at the end of “the lack of an image”

* Lines 214-216: Would the authors considering making two sentences here?

* Line 231: I would change “physician” for “doctor” to be consistent with the rest of the paper.

* Lines 257-206: Striking result which emphasizes the need to inform asylum seekers of their rights…

* Line 286: I would change the title of the sub-heading to “advantage and limitations of the TIS” as this section is not only about limits.

7. PLOS authors have the option to publish the peer review history of their article (what does this mean?). If published, this will include your full peer review and any attached files.

Reviewer #1: No

Reviewer #2: No

---

## [Author Response · Author response to Decision Letter 1]

5 Feb 2022

Thank you very much for your feedback on our work. Please find our responses to the reviewers’ comments in the "response to reviewers" attached file.

---

## [Editor Report · Decision Letter 2]

18 Feb 2022

Perception of the use of a telephone interpreting service during primary care consultations: A qualitative study with allophone migrants

PONE-D-21-18663R2

Dear Dr. CORMI,

We’re pleased to inform you that your manuscript has been judged scientifically suitable for publication and will be formally accepted for publication once it meets all outstanding technical requirements.

Kind regards,

Barbara Schouten

Academic Editor

PLOS ONE

---

## [Editor Report · Acceptance letter]

4 Mar 2022

PONE-D-21-18663R2 

Perception of the use of a telephone interpreting service during primary care consultations: A qualitative study with allophone migrants 

Dear Dr. Cormi:

I'm pleased to inform you that your manuscript has been deemed suitable for publication in PLOS ONE. Congratulations! Your manuscript is now with our production department. 

Kind regards, 

on behalf of

Dr. Barbara Schouten 

Academic Editor

PLOS ONE